# Next generation plasma proteome profiling to monitor health and disease

Wen Zhong [1], Fredrik Edfors [1], Anders Gummesson [2,3], Göran Bergström[2,4], Linn Fagerberg [1] & Mathias Uhlén [1,5 ✉]

The need for precision medicine approaches to monitor health and disease makes it important to develop sensitive and accurate assays for proteome profiles in blood. Here, we describe an approach for plasma profiling based on proximity extension assay combined with next generation sequencing. First, we analyze the variability of plasma profiles between and within healthy individuals in a longitudinal wellness study, including the influence of genetic variations on plasma levels. Second, we follow patients newly diagnosed with type 2 diabetes before and during therapeutic intervention using plasma proteome profiling. The studies show that healthy individuals have a unique and stable proteome profile and indicate that a panel of proteins could potentially be used for early diagnosis of diabetes, including stratification of patients with regards to response to metformin treatment. Although validation in larger cohorts is needed, the analysis demonstrates the usefulness of comprehensive plasma profiling for precision medicine efforts.

[1] Science for Life Laboratory, Department of Protein Science, KTH-Royal Institute of Technology, Stockholm, Sweden. [2] Department of Molecular and Clinical Medicine, Institute of Medicine, Sahlgrenska Academy, Gothenburg University, Gothenburg, Sweden. [3] Region Västra Götaland, Department of Clinical Genetics and Genomics, Sahlgrenska University Hospital, Gothenburg, Sweden. [4] Region Västra Götaland, Department of Clinical Physiology, Sahlgrenska University Hospital, Gothenburg, Sweden. [5] Department of Neuroscience, Karolinska Institutet, Stockholm, Sweden. ✉email: mathias.uhlen@scilifelab.se

One of the most important fields of biotechnology is the need to stratify patients to allow individualized treatment and monitoring of therapeutic interventions. This field called precision medicine has been dominated by genomics platforms, led by the rapid development in next generation sequencing. However, there is a need to move towards comprehensive proteome profiling of blood to take the next step in precision medicine, and this emphasizes the need for multiplex analysis of proteins in blood not only to understand the basis for wellness and disease, but also to facilitate precision medicine efforts aimed at early detection of disease, as well as stratification and monitoring of patients before and during therapeutic interventions[1,2]. The aim for such efforts is to identify signatures with pathophysiological importance forming an attractive bridge between genomes and phenotypes[3–6]. The objective is thus to probe the circulating plasma proteome of individuals with sensitive and specific assays that can allow massive sample throughput. However, progress has been hampered by the challenge to allow the quantification of thousands of proteins across more than a billion range in concentrations, starting with minute sample volumes.

Recently, proximity extension assay (PEA)[7] with qPCR read-out has been used to analyzed protein plasma levels in longitudinal studies[8–11] and genome association studies[12–14]. Here, we show that this analytical platform can be extended for simultaneous analysis of many more targets by the introduction of massive parallel sequencing, here referred to as PEA-NGS or Olink Explorer, without sacrifice on accuracy or sensitivity. This approach for next generation plasma profiling allows for simultaneous analysis of close to 1500 protein targets from minute blood samples (less than 3 mL of plasma). The method relies on four pillars allowing sensitive multiplex assays to be coupled with low cross-reactivity and minimal off-target events; (1) dual binding based on a specific monoclonal or polyclonal antibody recognizing more than one epitope on the target protein, (2) a proximity extension assay based on bringing two complementary DNA-barcodes in proximity by the antibody, (3) tag-sequences introduced into an amplicon to determine the sample origin and target protein, respectively, and (4) next generation sequencing to allow millions of amplicons to be sequenced and digitally counted. We use this method to study both health and disease with the objective to determine wellness profiles in individuals, as well as to identify protein markers for stratification and monitoring of patients during drug treatment.

## Results

### Multiplex protein analysis platform
The principle of the assay is summarized in Fig. 1. Antibodies having two different oligonucleotide tags are mixed and allowed to bind to the target protein (Fig. 1a). Upon binding to the same target, the complementary tags of each antibody anneal and allow for extension using a DNA polymerase (Fig. 1b, c). Each antibody and sample have a separate and unique barcode (Fig. 1d) and after amplification, a library of DNA fragments is prepared and sequenced (Fig. 1e). The number of reads corresponding to a particular antibody and samples is counted per sample barcode and can thereafter be translated into the original protein plasma concentration (Fig. 1f). In total, 1463 unique plasma proteins were analyzed using the technology (Supplementary Data 1), including 522 proteins not analyzed before using PEA. The Venn diagram (Fig. 1g) shows the overlap of proteins for the PEA-NGS assay as compared to the earlier described PEA-qPCR assay. Among them, 304 are actively secreted proteins in blood according to the recent annotation of the human secretome[15] (Supplementary Data 1).

### The study cohorts and clinical chemistry
The first cohort (wellness) consisted of 76 individuals recruited from the SCAPIS study[16], including 40 males and 36 females between 50 and 65 years of age. An inclusion criteria was the willingness to allow extensive sampling and physical examination every 3 months for 2 years. Extensive phenotype characterization of the subjects was conducted before the study to establish the inclusion and exclusion criteria for the definition of healthy subjects (see "Methods" section). The sample collection in combination with clinical chemistry analysis of 30 parameters as well as anthropometric measurements were conducted at the start of the study (visit 1), and after approximately 15–18 months (visit 2) and 21–24 months (visit 3) (Fig. 1h). The complete list of assessed clinical variables is available in Supplementary Data 2. The second cohort (type 2 diabetes) consisted of 48 individuals also recruited from the SCAPIS study, with no history of diabetes, and who were diagnosed during the population-based screening[17]. The sample collection in combination with clinical chemistry analysis of 30 parameters as described in the wellness study, anthropometric measurements as well as analysis of insulin, C-peptide, and the homeostatic model assessment of insulin resistance (HOMA-IR) were conducted at the start of the trial (visit 1) and after approximately 1 month (visit 2) and 3 months (visit 3) of treatment (Fig. 1h) (Supplementary Data 2).

### Comparison between the NGS and qPCR measurements generated with the proximity extension assay
To assess the performance and reproducibility of the NGS approach, we investigated the concordance between NGS and qPCR read-out. A set of 372 samples were analyzed using the conventional qPCR protocol and the result was compared to independent analysis of the same samples using the NGS protocol. Protein-wise Pearson correlation between NGS and qPCR platforms for each assay across samples was calculated, as shown in Fig. 2a. Most of the assays (~82%) were found to correlate well with a high cross-platform correlation >0.7. A pairwise correlation of all protein levels across the 372 samples also showed high concordance with a median Pearson correlation of 0.985 (Fig. 2b). No significant difference in correlation was observed between the two study cohorts (Supplementary Fig. 1b, c). Intra-platform and inter-platform variations have also been analyzed as exemplified by the interleukin-6 protein, which was run in four different NGS panels as well as the qPCR platform (Supplementary Fig. 1d). Both inter-platform and intra-platform correlations were high with an average Pearson correlation of 0.97, indicating high consistency of the measurements of protein levels using the PEA-NGS platform. Two examples of the correlation between the two methods are shown in Fig. 2c, d. The fibroblast growth factor 21 (FGF21) is an example of a protein with higher levels in the T2D patients[18,19], and the results from the two assays support this observation with consistent higher level of FGF21 in the T2D patients compared to the healthy individuals (Fig. 2c) with a correlation across all samples of 0.96 (Spearman) and 0.97 (Pearson), respectively. The second example is prokineticin 1 (PROK1) here shown (Fig. 2e, f) to have higher levels in males than females. The comparison between the two assays across all samples shows high correlation of 0.92 (Pearson and Spearman).

To identify all the proteins with sex-difference in plasma, an ANOVA analysis was performed with age and visit as covariates (Supplementary Data 3). Proteins with Benjamini–Hochberg adjusted P-value < 0.05 were considered as sex-related proteins. In total, 283 proteins were found to be significantly elevated in male samples, while 249 proteins showed significantly increased levels in female samples. A comparison of the fold changes of sex-related proteins in male and female samples between NGS and

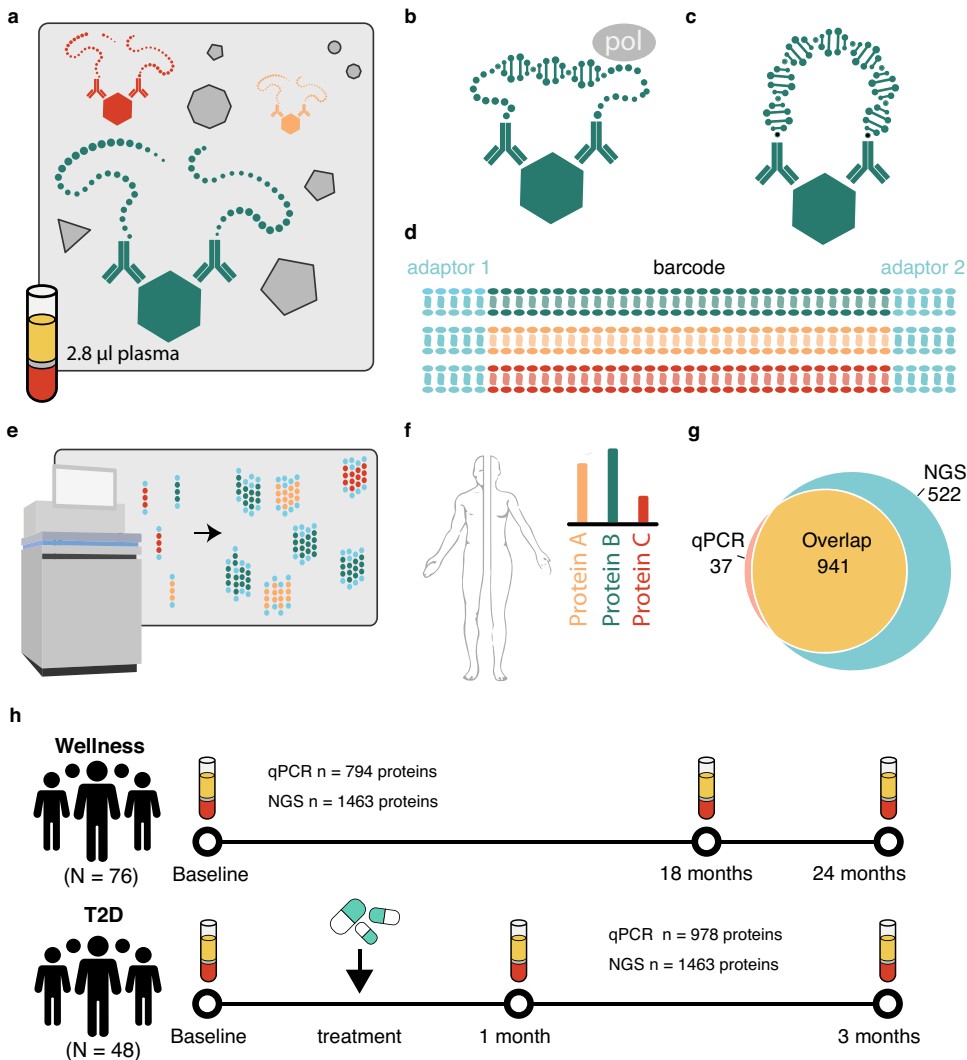

**Fig. 1 Principle of the PEA-NGS method and description of cohorts. a** DNA-conjugated antibodies bind to their target protein if present in the sample. **b** Antibodies with correct matched pairs can hybridize and are extended by DNA polymerase. **c** DNA-tags on correctly matched antibodies are extended and amplified through PCR amplification. **d** Unique barcode-regions identify each sample and protein and adaptors allow for NGS amplification and read-out. The color code indicates different barcodes. **e** NGS generates a digital signal, which translates to the number of DNA hybridization events that corresponds to the protein concentration in the original sample. **f** Relative protein concentration levels are obtained after normalization and quality control. **g** The number of protein targets in PEA-NGS as compared to the previous method based on qPCR assay. **h** Schematic description of the two analyzed cohorts.

qPCR measurements also shows a high consistence between these two platforms (Fig. 2g). Among them, 187 proteins were only identified using the PEA-NGS platform (Fig. 2h and Supplementary Data 3), as exemplified (Fig. 2i) by the neutrophic factor persephin (PSPN) and the enzyme pyrophosph/phosphodiesterase 2 (ENPP2), showing sex-specific differences in plasma with individual levels stable over the 2-year study period.

**Variance analysis of the plasma protein profiling and the genetic effects**. The approach described here allowed us to investigate ~700 proteins that have not been analyzed with PEA before. First, we analyzed the inter-individual and intra-individual variation of each protein in the longitudinal wellness cohort as described before[12] (Fig. 3a and Supplementary Data 4). We found that the majority of proteins have a higher variation between individuals (Fig. 3a) rather than within individuals, which is consistent with our previous findings[12]. These proteins are interesting to study in more depth in various disease cohorts. Second, we investigated the influence of genetic background on the individual plasma protein levels, using genome-

wide association analysis with 7.3 million variants identified by whole-genome sequencing, as previously described[12]. By introducing more proteins in the analysis, we found 331 significant associations ($P < 4 \times 10^{-11}$) between 143 proteins and 321 independent genetic variants (Supplementary Data 5). Among them, 69 proteins have not been identified before. Sentinel pQTL variant was determined as the variant with lowest P value at each pQTL locus and are visualized in Fig. 3b. In Fig. 3c, the PNLIPRP2 programmed cell death 6 (PDCD6), a lipase, which contributes to milk fat hydrolysis[20], is shown with the genetic variants associated with the plasma levels found in conjunction with the location of the protein-coding gene (cytoband 10q25.3 on the genome). The most significant association was found for variant rs4751995, which is a common truncation variant in PNLIPRP2 and was first described in 2003 as W357X in European cohort[21]. The longitudinal analysis during the three visits for the 76 subjects in the wellness study demonstrates that the protein levels are high for both the homozygote and heterozygotes for the variant, and the differences were stable during the 2-year study period.

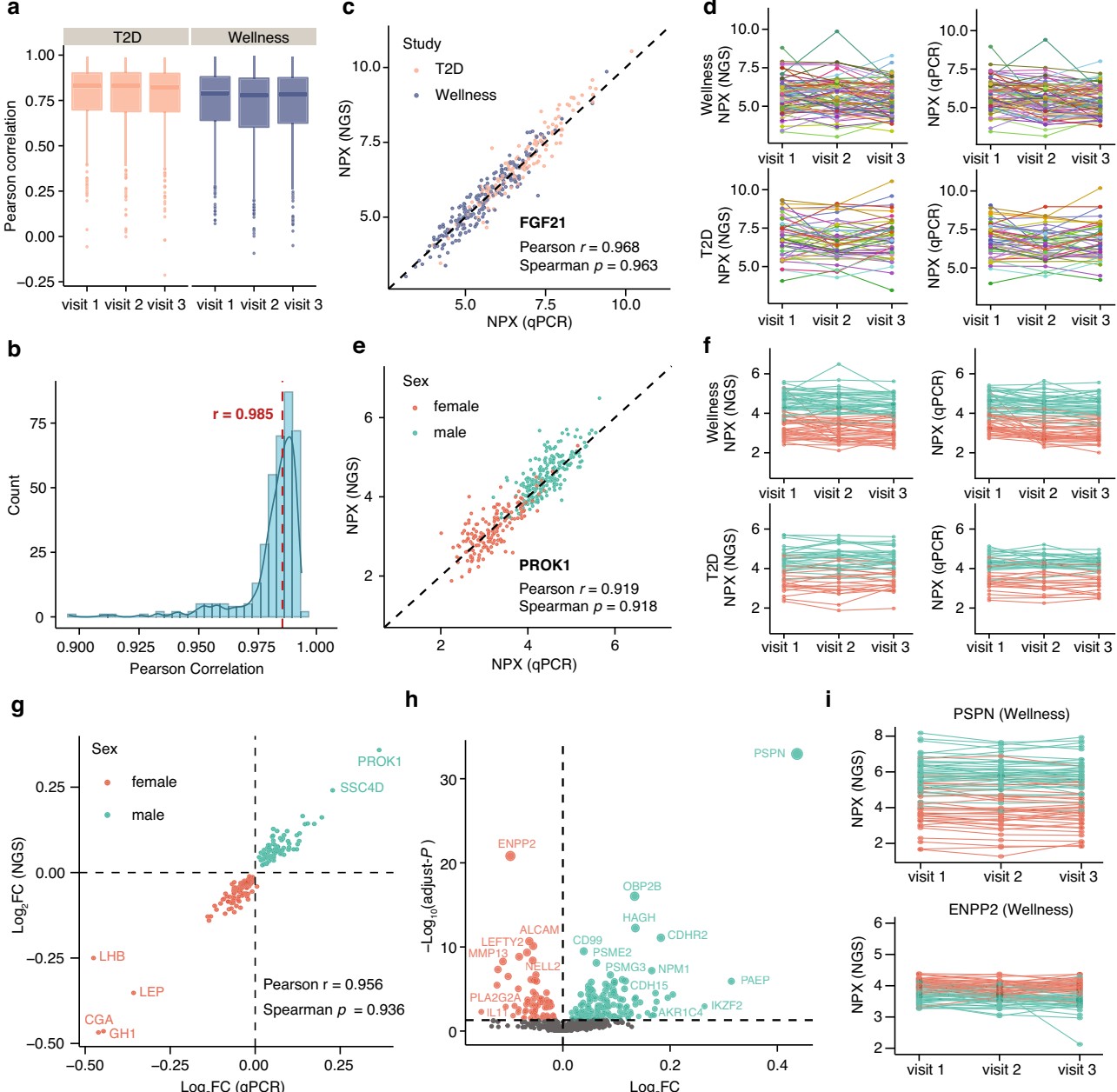

**Fig. 2 Correlation between PEA with NGS and qPCR measurements. a** Protein-wise correlation between NGS and qPCR for all proteins analyzed in each visit of the healthy and disease cohorts. Box plots show medians and the 25th and 75th percentiles, whiskers show the largest and smallest values ($n = 761$ proteins). **b** A combined boxplot and density plot showing the distribution of the sample-wise correlation values between NGS and qPCR measurements. The red dotted line shows the median Pearson correlation for all samples. **c** A scatter plot showing the pairwise correlation between the expression levels of NGS and qPCR platforms for the FGF21 protein. **d** The longitudinal protein profile for FGF21 in wellness and T2D. The color code indicates individuals. **e** A scatter plot showing the pairwise correlation between the expression levels of NGS and qPCR platforms for PROK1. **f** The longitudinal protein profile for PROK1 in wellness and T2D. The color code indicates individuals. **g** A scatter plot showing the pairwise correlation between the fold change in male and female samples measured by NGS and qPCR platforms for sex-related proteins identified in the wellness cohort. **h** A volcano plot showing the newly identified differentially expressed proteins in male and female samples. The $X$-axis represents $\log_2$ fold-change (FC) and the $Y$-axis represents $\log_{10}$ (adjusted $P$ values). Differentially expressed proteins were defined as proteins with adjusted $P$ values < 0.05 (three-way balanced ANOVA for gender with age and visit as covariates). Multiple test corrections have been applied for $P$-values using Benjamini and Hochberg method. **i** The protein concentration variation across visits one to three, with each individual connected with a dotted line for PSPN and ENPP2 proteins in the wellness cohort. The color code indicates females and males. Source data are provided as a Source Data file.

**Association of plasma protein levels with clinical measurements.** To compare the proteome profiles with anthropometrics and the clinical chemistry assays, an in-depth analysis of the relationship between plasma protein levels and clinical measurements was conducted using linear mixed-effect modeling and

the protein profiling from both wellness and T2D cohorts. In total, 919 of the proteins were found to be significantly associated with 42 clinical parameters (Supplementary Data 6). Among them, 1614 of the significant findings were related to 321 proteins measured here using the PEA-NGS assays. In addition, 53

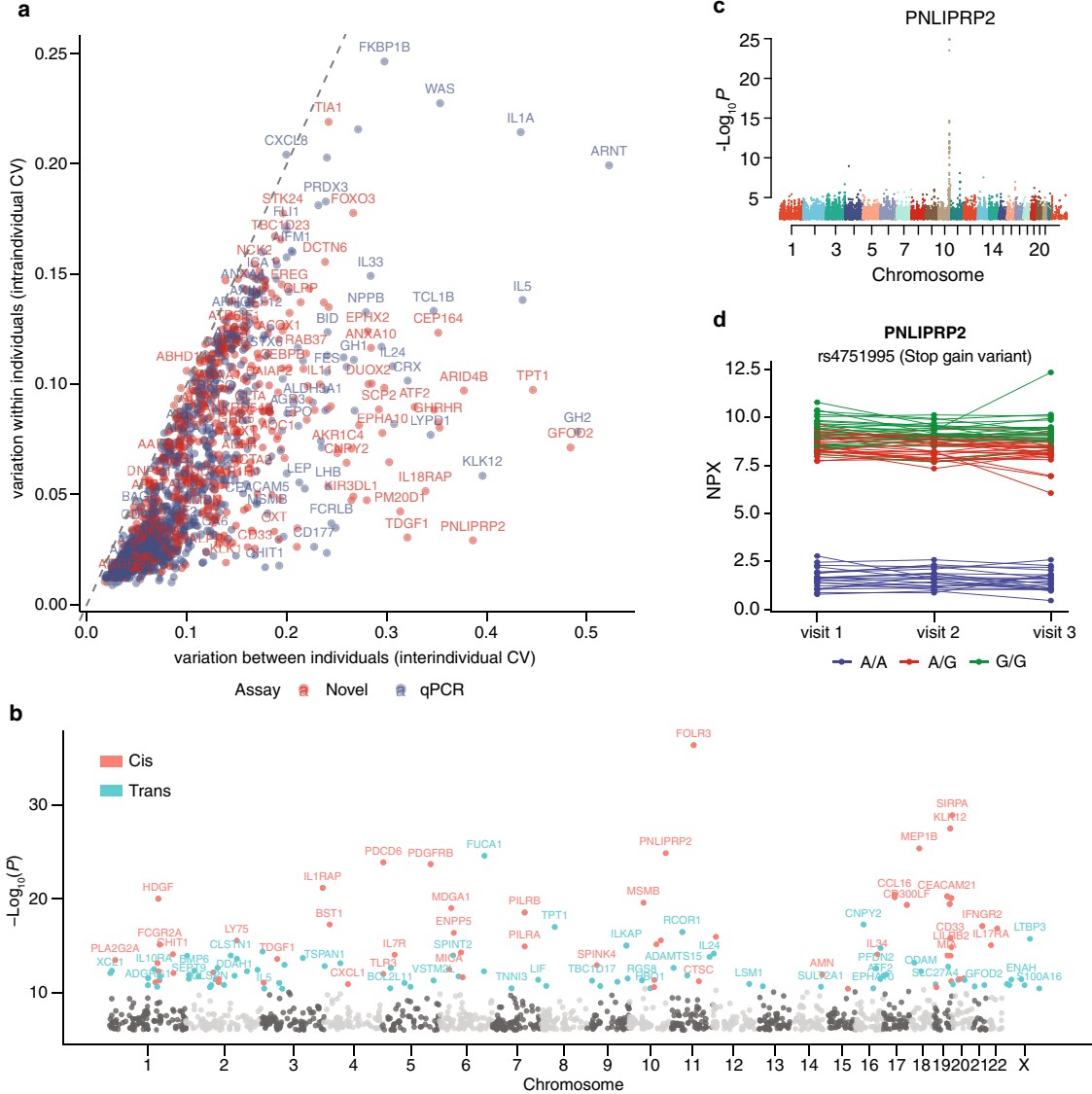

**Fig. 3 Variability of the plasma protein profiling and protein quantitative trait loci (pQTL). a** The inter-individual and intra-individual variation of protein levels calculated as the mean coefficient of variation (CV) for each protein within each visit and across all analyzed individuals ($n = 76$) in the wellness cohort, and as the mean CV for each protein within each individual across all visits ($n = 3$), respectively. **b** Manhattan plots of the sentinel pQTL per protein visualizing the cis-pQTLs and trans-pQTLs for all proteins with significant associations. The light and dark grey dots represent the not significant associations split into chromosome location. **c** Manhattan plot of protein PNLIPRP2 shows the genome locations of all associated pQTLs. **d** The longitudinal protein concentration across visits one to three with each individual connected with a dotted line for PNLIPRP2, colored by the different variants. Source data are provided as a Source Data file.

proteins were also identified with significant genetic associations (Fig. 4a) suggesting that a large contribution to the plasma levels of these proteins during adult life is determined by genetic factors. In Fig. 4b, the proteins most strongly associated with clinical assays based on both wellness and T2D cohorts are shown, here restricted to the proteins exclusive for the PEA-NGS assay (see full list in Supplementary Data 7). Many proteins related to sex, BMI, lipid, and immune response were identified, as well as proteins involved in liver and kidney function. These are interesting targets to study more in-depth in larger cohorts to investigate their potential role for individualized stratification of health and disease.

**Identification of individuals with high risk for T2D.** An interesting topic in the diabetes field is to be able to stratify T2D patients into different categories based on clinical and molecular

parameters[22]. As shown in Fig. 5a, the T2D cohort can roughly be stratified into two groups with BMI larger or smaller than 30 at baseline. These groups, here called obesity (>30) and non-obesity (<30), are roughly equal in size, 28 (obesity) and 20 (non-obesity). In contrast, the majority of individuals in the wellness study are with BMI under 30. First, we investigated the plasma proteome profile difference between the two T2D cohorts. Four proteins were found to associate with the BMI-difference among the T2D patients, as exemplified by the famous obesity marker leptin (Supplementary Fig. 3). We next investigated the difference in protein profiles between the T2D non-obesity group and the non-obese healthy (wellness) control group. The analysis identified a number of proteins ($n = 32$) differing between the healthy and T2D individuals, as shown in Fig. 5b and Supplementary Data 8, with some examples of proteins with decreased (Fig. 5c) or increased (Fig. 5d) plasma levels in the T2D patients at baseline.

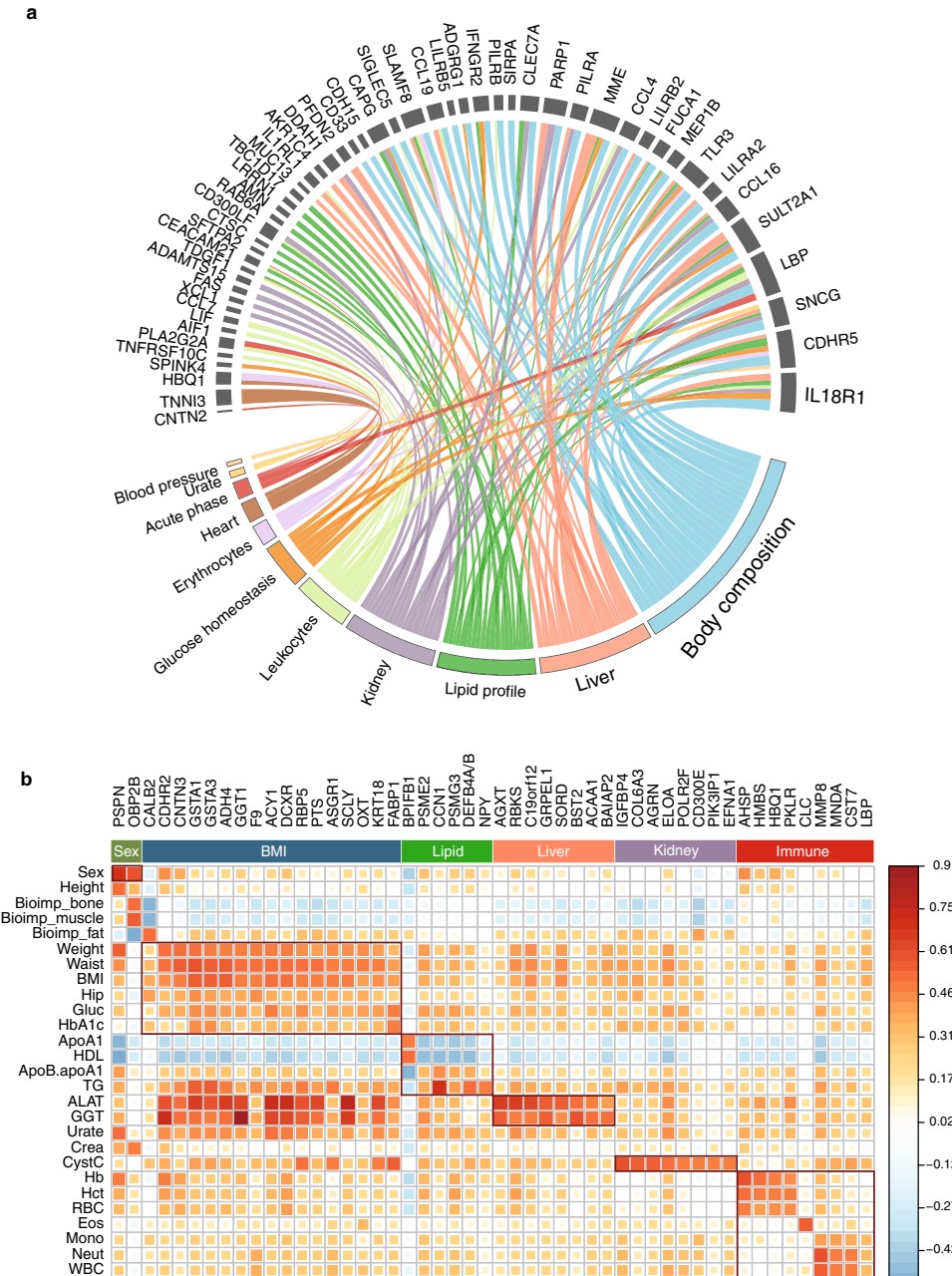

**Fig. 4 Associations of proteins with clinical assays. a** Chord diagram presenting the 53 proteins with pQTLs related to clinical measurements. The size of the link is defined as the absolute value of coefficient of the corresponding effect (mixed-effect modeling analysis for each protein and clinical measurement adjusted for gender, age and visit). **b** A heatmap showing the top 50 proteins most strongly associated with clinical assays based on both wellness and T2D cohorts. Only proteins who are exclusive in the PEA-NGS are listed. See complete results of the pairwise Pearson correlation values between all proteins and clinical assays in Supplementary Data 7. Source data are provided as a Source Data file.

These proteins are thus potential plasma protein biomarkers for identifying individuals with high risk for T2D in population screens independent on BMI. A principle component analysis (PCA) based on the 32 differentially expressed proteins shows that the T2D patients clustered together, regardless of anthropometrics parameters (BMI), and separate from the healthy individuals, using the combined protein profile of this panel of 32 proteins.

**Stratification of T2D patients with regards to response to metformin treatment.** Among the T2D patients, 40 individuals were treated with metformin and here we stratified these into

three groups based on the response of therapeutic intervention, as determined by the decreased level of fasting plasma glucose (FPG) after three months treatment (Fig. 6a). The groups included responders (FPG decrease >1), non-responders (FPG decrease <0.1) and an indeterminate group (FPG decrease between 0.1 and 1). Of note, one patient with large and opposite changes of FPG in visit 2 (−2.8) and visit 3 (1.6) was also grouped into the indeterminate group. Changes of plasma levels during diabetes treatment as well as the comparison with healthy individuals in the top 50 most highly associated proteins with fasting glucose are present in Supplementary Fig. 4. As shown, the treatment results in partial recovery towards normal protein

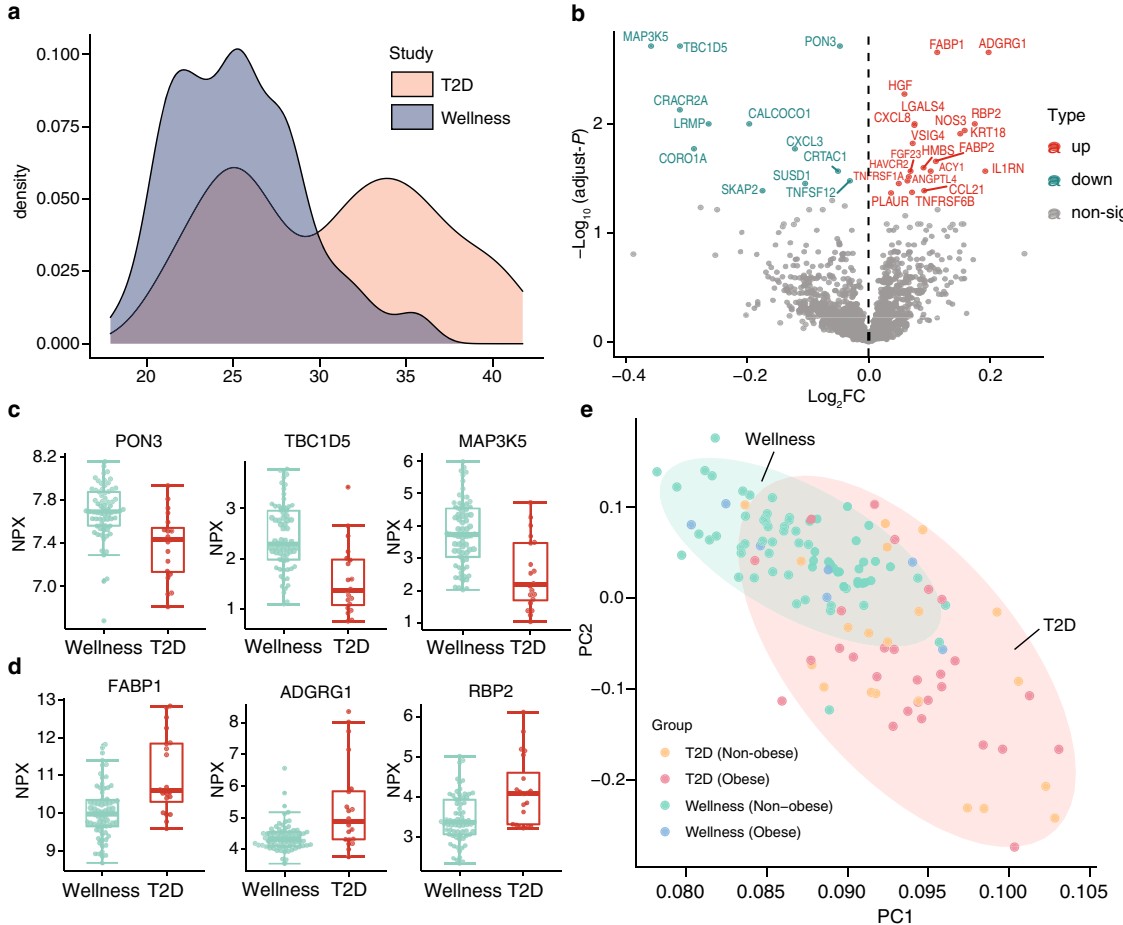

**Fig. 5 Plasma proteins associated with type 2 diabetes. a** The distribution of body mass index (BMI) in the two cohorts. **b** A volcano plot showing the differentially expressed proteins at baseline in the non-obesity group in T2D and wellness cohorts. The X-axis represents $\log_2$ fold-change (FC) and the Y-axis represents $\log_{10}$ (adjusted P-values). Differentially expressed proteins were defined as proteins with adjusted P-values < 0.05 (two-way balanced ANOVA for study cohort with gender as covariate). Multiple test corrections have been applied for P-values using Benjamini and Hochberg method. **c** Three examples of the upregulated proteins in the non-obesity T2D patients compared to the non-obesity healthy individuals in the wellness cohort. Box plots show medians and the 25th and 75th percentiles, whiskers show the largest and smallest values ($n = 89$ samples), error bars represent mean ± SD. **d** Three examples of the down-regulated proteins in the non-obesity T2D patients. **e** PCA results presenting the expression patterns of healthy and disease individuals at baseline based on the protein profiles of differentially expressed proteins between the non-obesity groups in T2D and wellness cohorts. Source data are provided as a Source Data file.

levels for most of the proteins and the effect is more pronounced in the responder group as compared to the non-responder group for many of the proteins.

To explore if the plasma protein levels at baseline were associated with the response to metformin treatment, a mixed-effect modeling analysis was conducted (Supplementary Data 9). Median levels of the top 30 most highly associated proteins with metformin treatment response are present in Fig. 6b. As shown, different protein expression patterns between responders and non-responders could be observed with a majority of proteins showing higher plasma levels at visit 1 (before start of treatment) in the responder group, but also some proteins (GPA33, LEP, TPSAB1, MZB1, GALNT10, and IFNLR1) with lower plasma levels as compared to the non-responders. In Fig. 6c, the protein levels in the three groups are shown as boxplots for four selected examples. Unsupervised clustering analysis was then performed on the protein profiles of the top 30 proteins at baseline for all T2D patients with metformin treatment (Fig. 6d). Interestingly, all non-responders were clustered together and distinct from the responders, indicating that this panel of proteins can be used to predict if the metformin treatment for a particular patient will be successful or not before the start of the therapeutic intervention.

## Discussion

Here, we describe a study to use a sensitive and accurate multi-plex analysis of blood proteins based on proximity extension assay combined with next generation sequencing read-out. Close to 1500 human proteins were assayed allowing minute (micro-liter) amounts of blood to be analyzed with simultaneous analysis of low abundant proteins presented in concentrations more than a billion-fold lower compared to the most abundant proteins in human blood. The Olink Explorer method described here thus complement the pioneering Slow-Off rate Modified Aptamer (SOMAmer) technology[23,24] used for several population-based studies[25–27], and enables an antibody-based method combining sensitivity and parallel plasma analysis of thousands of protein targets.

The results presented here support earlier observations[8,12] that each individual has a unique blood protein fingerprint with larger inter-individual variations as compared to the intra-individual variation. Several proteins with strong association with known clinical parameters have here been identified and these are interesting to study in larger cohorts to validate them as clinical markers in routine settings. The genome association studies reported here also support earlier observations that many plasma

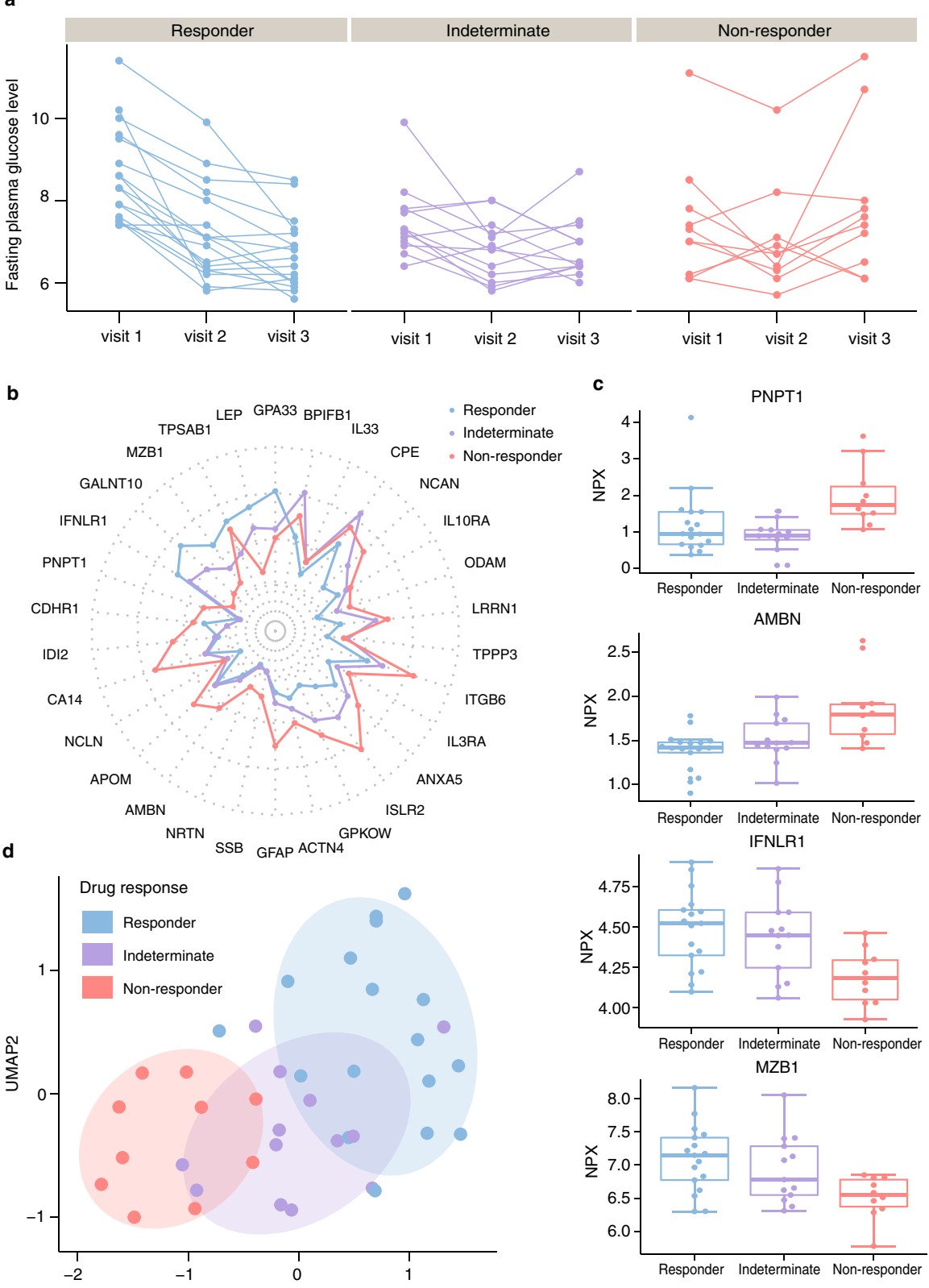

**Fig. 6 Plasma proteins associated with the metformin treatment response. a** The longitudinal fasting plasma glucose distribution across the three visits in T2D patients with metformin treatments. The color code indicates three different drug response groups (responder, green; indeterminate, purple; non-responder, orange). **b** Radar plot showing median levels of top 30 proteins associated with metformin treatment response in three groups (see expression patterns of all identified proteins in Supplementary Fig. 5). **c** Four examples of the distribution of the plasma levels of the most significant proteins at baseline in the three drug response groups. Box plots show medians and the 25th and 75th percentiles, whiskers show the largest and smallest values ($n =$ 40 samples), error bars represent mean ± SD. **d** A UMAP analysis of the expression patterns of T2D patients in the three drug response groups at baseline using the top 30 associated proteins. Source data are provided as a Source Data file.

protein levels in adult life are determined at birth by genetics[12], and 69 genetic variants (pQTLs) not previously described were identified using the multiplex assay. This demonstrates that genetics should be taken into account when assessing an individual´s plasma protein levels in population studies.

The analysis of the T2D patients revealed many interesting observations. The characteristics of T2D and its association with the epidemic of obesity have been extensively studied, but relatively little is known about the incidence of lean diabetes and the progression of disease during treatment[28]. The results presented here support the notion that broad biochemical alterations are present already at the onset of type 2 diabetes and that protein profiling could deliver individualized health assessments of cardiometabolic diseases[4,17]. The anthropometrical parameters of the T2D cohort allowed us to stratify the patients into two distinct groups based on BMI and an analysis of the protein profiles between the health individuals and the T2D patients at base line (visit 1) revealed several proteins, that could be useful for early detection of disease based solely on protein profiles. Interestingly, the proteome profile showed the possibility to stratify the patients into responders and non-responders of naïve metformin drug treatment and analysis with a selected panel of proteins demonstrated that persons less likely to respond to treatment could be identified before the start of treatment. The suggestion that a panel of protein assays could be used to guide the physician regarding choice of drug treatment is highly promising, but it is important to point out that more in-depth analysis of these plasma protein profiles must be performed in larger disease cohorts to validate their use as clinical biomarkers.

The method described here has opened up the possibility to perform next generation plasma proteome profiling to allow massive screening of various patient cohorts to provide plasma profiles of relevance for health and disease. This technology allows analysis of minute samples over the challenging dynamic range of plasma proteins, which will be highly beneficial for precision medicine efforts. In summary, we describe an approach suitable for comprehensive protein profiling, taking advantage from the observation that each individual has a unique protein profile and many proteins change during disease allowing stratification and monitoring of patients during treatment.

## Methods

**The wellness profiling study**. The Swedish SciLifeLab SCAPIS Wellness Profiling (S3WP) program is non-interventional with the aim to collect longitudinal clinical and molecular data in a community-based cohort, as previously described[8–10,12]. The study consists of 101 healthy individuals recruited from the Swedish CArdioPulmonary bioImage Study (SCAPIS), which is a prospective observational study with 30,154 individuals enrolled at ages between 50 and 64 years from a random sampling of the general Swedish population[16], from 2015 to 2018. Hence, all subjects had been extensively phenotyped in SCAPIS before entering the S3WP study. For inclusion in the study subjects must fulfil the following criteria: (1) signed informed consent to participate in the study, (2) randomly selected and included in the Gothenburg SCAPIS cohort, (3) 50 but not yet 65 years of age at the time of selection from the SCAPIS cohort, (4) Ability to understand instructions and complete questionnaires, as judged by the study staff. The exclusion criteria in the S3WP study included: (1) previously received health care for myocardial infarction, stroke, peripheral artery disease, or diabetes, (2) presence of any clinically significant disease which, in the opinion of the investigator, may interfere with the results or the subject's ability to participate in the study, (3) any major surgical procedure or trauma within 4 weeks of the first study visit, or (4) medication for hypertension or hyperlipidemia. Examinations in combination of sample collection (blood, urine and feces) were performed every third month (±2 weeks) in the first year and approximately a 6-month interval in the second year. During the study, 99 subjects completed the first year and 94 completed the second year. The study is approved by the Ethical Review Board of Göteborg, Sweden (registration number 407-15). All participants provided written informed consent. The study protocol conforms to the ethical guidelines of the 1975 Declaration of Helsinki (Supplementary Note 1 and 2).

**The type 2 diabetes (T2D) study**. The T2D study is an extension of the S3WP study with the aim to perform molecular phenotyping of T2D before and after diabetes treatment[17]. Fifty-two subjects at age between 50 and 65 years with no history of diabetes and diagnosed during population-based screening examinations were enrolled from the Sahlgrenska University Hospital, Gothenburg, from 2016 to 2018. The diagnosis of diabetes was based on the Swedish standard, corresponding to the American Diabetes Association standards (1), and subjects who met diabetes criteria were scheduled for a second glucose measurement on a separate occasion and enrolled if diabetes diagnosis was confirmed. For inclusion in the study subjects must fulfil the following criteria: (1) signed informed consent to participate in the study, (2) age 50–65 years, (3) T2D diagnosis (the combination of either fasting p-glucose ≥7.0 mmol/L or an oral glucose tolerance test 2 h p-glucose ≥11.1 mmol/L [or ≥12.2 mmol/L if measured capillary] at two separate occasions). The exclusion criteria in the T2D study included: (1) diabetes medication before study start, (2) severe hyperglycemia requiring hospitalization or immediate insulin treatment, as judged by the investigator, (3) presence of any clinically significant disease which, in the opinion of the investigator, may interfere with the subject's ability to participate in the study, (4) any major surgical procedure or trauma within 4 weeks of the first study visit. Examinations were performed at baseline and after 1 and 3 months of guideline-based diabetes treatment. The T2D study is observational and diabetes treatment was part of standard clinical care according to first-line therapy with lifestyle change including weight management and physical activity, with or without metformin as judged by the treating physician. Complete samples from all visits were obtained from 48 subjects, including 29 males and 19 females. The study is approved by the Ethical Review Board of Göteborg, Sweden (registration number 448-16). All participants provided written informed consent. The study protocol conforms to the ethical guidelines of the 1975 Declaration of Helsinki (Supplementary Note 1 and 3).

**Examinations and questionnaires**. All visits in the wellness study and the T2D study were performed using the same protocol. All subjects were fasting overnight (at least 8 h) before the visits. Physical examinations included height, body weight, waist and hip circumference, body fat using bioelectrical impedance (Tanita MC-780MA) and blood pressure (Omron P10). The body mass index (BMI) was calculated by dividing the weight (kg) by the square of the height (m). A selection of questions from the initial SCAPIS questionnaire was repeated to note any changes in health and lifestyle factors between each visit.

**Clinical chemistry and hematology measurements**. Clinical chemistry and hematology measurements included fasting glucose, haemoglobin A1c (HbA1c), triglycerides (TG), total cholesterol, low-density lipoprotein (LDL), high-density lipoprotein (HDL), apolipoprotein A1 (ApoA1), apolipoprotein B (ApoB), ApoA1/B ratio, creatinine, high sensitive C-reactive protein (hsCRP), alanine aminotransferase (ALAT), gamma-glutamyltransferase (GGT), urate, cystatin C, troponin T (TNT), N-terminal pro-brain natriuretic peptide (NT-proBNP), haemoglobin (Hb), white blood cell count (WBC), red blood cell count (RBC) and platelet count. In addition, insulin and C-peptide was measured in the diabetes group and the homeostatic model assessment of insulin resistance (HOMA-IR) was calculated according to the formula: fasting insulin (mU/L) × fasting glucose (mmol/L) / 22.5[29]. In total, a variety of 33 clinical chemistry parameters were included in the study, see more details in Supplementary Data 2.

**Plasma sample collection and preparation**. Plasma samples were collected after an overnight fast and at the same visit as the clinical examinations, using EDTA sample tubes using venipuncture protocols, and stored at −80 °C until analysis. All available plasma samples were analyzed using PEA-qPCR as previously described[8,12]. For PEA-NGS analysis, the sample size was decided based on availability of analysis capacity, and each individual is analyzed three times. We randomly selected 76 subjects with full longitudinal data and plasma samples at the start of the study (visit 1) and after approximately 15–18 months (visit 2) and 21–24 months (visit 3) in the wellness study, and 48 subjects with complete series of plasma samples in the T2D study.

**Plasma protein profiling using next generation sequencing read-out**. Specific antibodies targeting 1472 proteins are each labelled separately with unique PEA oligonucleotide probes, each antibody is labeled with two separate and complementary sequences. The conjugated antibodies are mixed into four separate 384-plex panels (372 proteins and 12 internal controls used for QC and normalization, see below) focused on inflammation, oncology, cardiometabolic and neurology proteins respectively (see full list in Supplementary Data 1). Each of the four 384-plex panels contain three control assays (interleukin-6 (IL6), interleukin-8 (CXCL8), and tumor necrosis factor (TNF) used for quality control (QC). The analytical performance of each of the protein assays included in the panel is validated based on specificity, sensitivity, dynamic range, precision, scalability, and detectability in both healthy and pathological plasma and serum samples. Briefly, samples were randomized (different samples from the same individual and matched case–controls were present within the same plate) and 2.8 µL of plasma were incubated overnight with antibodies conjugated to PEA probes at +4 °C. Following the immune reaction, a combined extension and pre-amplification mix were added to the incubated samples at room temperature for PCR amplification. The PCR amplicons were thereafter pooled before a second

PCR amplification step was performed with additions of individual sample index sequences. After pooling of samples, bead purification and QC of the generated libraries were followed on a Bioanalyzer. Finally, the sequencing were carried out using Illumina's NovaSeq 6000 instrument using two S1 flow cells with $2 \times 50$ base read lengths. Counts of known barcode sequences were thereafter translated into normalized protein expression (NPX) units through a QC and normalization process. NPX is a relative protein quantification unit and values are reported on a log2 scale. The values are calculated from the number of matched counts on the NovaSeq run. Data generation of NPX consists of three main steps: normalization to the extension control (known standard), log2-transformation, and level adjustment using the plate control (plasma sample). Specifically engineered internal controls were added to each sample and are utilized to reduce intra-assay variability. These include one immuno-based control (incubation step) using a non-human assay, one extension control (extension step) composed of an antibody coupled to a unique DNA-pair always in proximity and, also, one amplification control (amplification step) based on a double stranded DNA amplicon. In addition, each sample plate includes sample controls used to estimate the precision (intra-CVs and inter-CVs). Three negative controls (buffer only) are utilized to set background levels and calculate limit of detection (LOD), three plate controls (plasma pool) adjust levels between plates (thus improving inter-assay precision, allowing for optimal comparison of data derived from multiple runs), and finally two sample controls (reference plasma) are included to estimate precision. The PEA-NGS analysis was conducted using the Olink Explore strategy conducted at Olink Proteomics.

**Plasma protein profiling with qPCR read-out.** As previously described[7,8,12], 1 μL sample buffer (PBS with 0.1% BSA), antigen-spiked buffer, or biological sample) was mixed with 0.3 μL of each proximity probe mix (A and B), 0.3 μL Incubation Stabilizer (Olink Proteomics, Uppsala, Sweden) and 2.1 μL Incubation Solution (Olink Proteomics) and incubated overnight at 4 °C. A combined extension and preamplification mix (96 μL) containing 10 μL MUX PEA Solution (Olink Proteomics), 0.5 units Pwo (DNA Gdansk, Poland), 1 μM forward + reverse universal preamplification primers, and 1 unit hot-start DNA polymerase was added to each reaction at RT. After mixing and a total 5-min incubation, the plate was transferred to a thermocycler running an initial extension step to unite the two oligonucleotides (50 °C, 20 min), immediately followed by a hot-start step (95 °C, 5 min) and 17 cycles of amplification (95 °C, 30 s; 54 °C, 1 min; and 60 °C, 1 min). Amplification was performed with universal flanking primers to amplify all 96 sequences in parallel. Finally, 2.8 μL of the preamplification products were mixed with 7.2 μL buffer containing 5 μL MUX Detection Solution (Olink Proteomics), 0.071 units Uracil-DNA glycosylase (DNA Gdansk) used to digest the DNA templates and remaining universal primers, and 0.14 units hot-start polymerase. Five microliter from the sample mix above were transferred to the sample inlet wells of a microfluidic real-time PCR chip (96.96 Dynamic Array IFC, Fluidigm Biomark). Five microliter from respective well of an Assay Plate (Olink Proteomics) containing 9 μM sequence-specific internal detection primers, 2.5 μM molecular beacon in 1× DA Assay Loading Reagent (Fluidigm) were transferred to the assay inlet wells). The chip was run in a Biomark instrument with the following program: Thermal mix (50 °C, 2 min; 70 °C, 30 min; and 25 °C; 10 min), Hot-start (95 °C, 5 min), PCR Cycle 40 cycles (95 °C, 15 s and 60 °C, 1 min) according to the manufacturer's guidelines (http://www.fluidigm.com/biomark-hd-system.html).

To minimize inter-run and intra-run variation, the samples were randomized across plates and normalized using both an internal control (extension control) and an inter-plate control, and then transformed using a pre-determined correction factor. The pre-processed data were provided in the arbitrary unit Normalized Protein eXpression (NPX) on a log2 scale and a high NPX presents high protein concentration. In this study, eleven Olink panels have been used including Cardiometabolic, Cell Regulation, Cardiovascular II (CVD II), Cardiovascular III (CVD III), Development, Immune Response, Oncology II, Inflammation, Metabolism, Neurology, and Organ Damage. Quality control (QC) was performed at both sample and protein levels. A sample will flag (not pass the QC) if incubation control deviates more than a pre-determined value (+/− 0.3) from the median value of all samples on the plate (www.olink.com). To reduce the batch effect between samples run at different times, bridging reference samples from different visits were also run on plates from the different batches. Reference sample normalization based on bridging samples was conducted to minimize technical variation between batches (www.olink.com).

**Comparison of the plasma protein profiling with NGS and qPCR read-out.** Intensity normalization was carried out for each protein to reduce the technical variation between the two methods to ensure the median value for each protein target is the same in different batches without changing the relative protein levels across samples (www.olink.com). Briefly, for each protein, the overall median value for all samples and two platforms was firstly calculated. Then, for each protein and platform, the platform specific median value was also calculated. For each protein, the normalized value for each sample and each platform is the subtraction of platform specific median value from the NPX value with the addition of overall median value. Two examples of the levels of proteins before and after normalization in NGS and qPCR platforms are shown in Supplementary Fig. 1a.

**Whole genome sequencing.** Genomic DNA was quantified using Qubit 2.0 Fluorometer (Invitrogen), fragmented into average 350-bp fragments using E220 focused-ultrasound sonicator (Covaris), and 1 μg of fragmented DNA was converted into sequencing ready library using TruSeq DNA PCR-free HT Sample preparation method (Illumina). The obtained library was quantified using KAPA SYBR FAST qPCR (Kapa Biosystems) and pair-end ($2 \times 150$ bp) sequenced to average 30× coverage on the HiSeq X system (Illumina) using v2 flowcells. Demultiplexing was done without allowing any missmatches in the index sequences. Bioinformatic analysis of the sequence data was carried out using Mutation Identification Pipeline (version 4.0.18)[30]. Briefly, alignment was done using BWAmem (v0.7.17) using reference genome GRCh38.p7, and single-nucleotide and insertion/deletion variants called using GATK best practices pipeline (https://software.broadinstitute.org/gatk/best-practices, GATK v3.6). Structural variants were called using Manta (v1.0.3)[31]. Variants in the any of the 56 ACMG genes[32] were excluded from further analysis in order to avoid secondary findings.

The VCF files were then converted to PLINK-format with the PLINK software, version 19[33]. Quality control (QC) was conducted to avoid false findings, as described by Zhong et al.[12]. The exclusion criteria for variants include: (1) remove individuals with high missing genotype rates (>5%); (2) remove SNPs fail the genotyping rate threshold 0.05; (3) remove SNPs with low minor allele frequencies (MAF) (<5%); (4) remove SNPs fail the Hardy–Weinberg equilibrium (HWE) test ($P < 0.001$). In total, 7,275,131 high-quality variants were identified in all samples.

**Genome-wide association analysis.** Baseline protein concentration level for each subject in the wellness study was calculated as a median of NPX values across three visits. Association between each protein and genetic variant was performed using a linear regression model adjusted for age and sex at baseline using PLINK v1.9[33]. No significant association between protein levels and ancestry was observed by using mixed effect modeling in the study. Therefore, no correction for ancestry was applied. Bonferroni-adjusted $P$-value < $4 \times 10^{-11}$ (genome-wide threshold of $P = 5 \times 10^{-8}$, 1463 proteins tested) were considered to be significant in the study. Functional annotation of variants was performed using Ensembl Variant Effect Predictor (VEP) v87[34]. A cis-pQTL variant was defined as a SNP residing within 1 megabase (Mb) upstream or downstream of the transcription start site of the corresponding protein-coding gene. A SNP located >1 Mb upstream or downstream of the gene transcript or on a different chromosome from its associated gene was categorized as a trans-pQTL variant. Linkage disequilibrium (LD) was computed as the square of Pearson's correlation ($r^2$) between genotype allele counts across 101 subjects. To identify independent pQTLs for a given protein, LD $r^2 > 0.1$ with window size 1 Mb was first used to exclude the correlated variants. For proteins with multiple pQTLs, a conditional analysis was then carried out in which the genetic associations were re-calculated using the sentinel SNP as covariate. Only associations with conditional $P$-value < 0.01 were considered to be independent pQTLs.

**Statistical analysis and visualization.** Data analysis and visualization was performed using R (v3.6.3)[35] with the tidyverse suite of R packages[36] and the ggplot2 R package[37]. Mixed-effect modelling was performed using the lme4 package[38] and Kenward-Roger approximation[39] was used to calculate $P$-values which were subsequently adjusted for multiple testing using Benjamini–Hochberg method[40]. $P$-values were considered significant if less than 0.01. Differential expression analysis was carried out using analysis of variance (ANOVA) method with the built-in R function anova(). False discovery rate (FDR) was calculated by using p.adjust() function in R, which uses Benjamini−Hochberg method. Proteins with FDRs < 0.05 were considered differentially expressed. PCA and Uniform Manifold Approximation and Projection (UMAP) have been performed based on scaled NPX values using the R packages pcaMethods[41] and umap[42]. Chord diagram and radar chart were generated using R packages circlize[43] and fsmb[44].

**Reporting summary.** Further information on research design is available in the Nature Research Reporting Summary linked to this article.

## Data availability

All summary statistics and association data are available in the supplementary material. The participant-level genotype and phenotype datasets of S3WP program, including genetic mutations, plasma protein profiling with both NGS and qPCR readout, clinical chemistry, anthropometric measurements and questionnaires, have been deposited with the Swedish National Data Service (https://snd.gu.se/sv/catalogue/study/preview/88efa94d-39b3-4a50-8b3b-87b1abedefd4, a data repository certified by Core Trust Seal). Due to patient consent and confidentiality agreements, the dataset can only be made available for validation purposes by contacting snd@snd.gu.se. Data access will be evaluated according to Swedish legislation. Data access for research related questions in the S3WP program can be made available by contacting the corresponding author. Source data are provided with this paper.

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

## Acknowledgements

We are grateful to Olink Proteomics AB for providing the technology for PEA-NGS and conducting the assays as part of a strategic technology development program. We acknowledge the entire staff of the Human Protein Atlas program and the Science for Life Laboratory (SciLifeLab) for their valuable contributions. We also acknowledge the study staff at the Wallenberg Laboratory, Department of Molecular and Clinical Medicine, for their excellent and dedicated management of recruitment, clinical examinations and handling of samples. The processing of whole genome sequencing data was performed on resources provided by SNIC through Uppsala Multidisciplinary Center for Advanced Computational Science (UPPMAX). Main funding was provided from the Erling Persson Foundation (KCAP).

## Author contributions

M.U. conceived and designed the analysis. W.Z., F.E., L.F., G.B., and A.G. collected and contributed data to the study. W.Z. and M.U. performed the data analysis. A.G. and G.B. supplied clinical material. W.Z. and M.U. drafted the manuscript. All authors read and approved the final manuscript.

## Funding

## Competing interests

The authors declare no competing interests.
