## [Peer Review File · Nature Communications]

REVIEWER COMMENTS

Reviewer #1 (Remarks to the Author):

Zhong et al use the latest Olink technology with a nucleic acid readout to profile ~1500 plasma proteins from their wellness cohort. This platform enables the probing of 522 proteins more than their recently published study that used the PCR readout. They extend their previous observations by finding more sex specific proteins and more genetic associations. They also have preliminary differences between T2D responders and nonresponders

The strengths of the study are:

- 1) The authors demonstrate that the new readout format is highly reproducible and similar to the PCR readout.
- 2) The increase the amount of sex specific proteins and loci associated with protein abundance
- 3) The preliminary study identifying a set of proteins associated in T2D associated with metformin responders and nonresponders.
- 4) The technical execution of the study is fine.

The weakness of the study are:

- 1) The authors did not invent the technology—they are simply using it as are many other labs.
- 2) There is no increase concepts from their very recently published wellness paper and a very modest increase in information
- 3) The T2D could be taken much further.

Over the advance feels modest.

Reviewer #2 (Remarks to the Author):

The paper presents an impressive and relevant approach “next generation plasma profiling”, where large parts of the setup have been published previously. I will limit my comments to the dry genetics data part reflecting the influence of genetic variations on plasma levels, as I am not fully able to assess the details in the experimental part. As far as I understand this slice of the workflow, the starting point is 30X WGS data. I do not really understand how the data are leveraged, or rather possibly not leveraged.

Following the NGS data generation the possibly exciting variants seem to be ignored and a suboptimal GWAS ($n=101$) for variants with $MAF>5\%$ is carried out. It is surprising why the GWAS made in the first place when 30X WGS data are available. The logic behind this should be explained better, or modifications should be made. There are also no sign of using the genetics to adjust for population structure and cryptic relatedness that would be ideal here. As this aspects is not part of what is previously published it would be good to explain the logic behind this strategy.

Reviewer #3 (Remarks to the Author):

This paper provides a thorough description of a very interesting novel approach for comprehensive and sensitive plasma profiling based on proximity extension assay combined with next generation sequencing. The manuscript is well structured and well written overall.

Major points:

Most of the methods and results are clearly presented and interpreted. However, I am concerned that the claim prominently expressed in the abstract and elsewhere that „The analysis also allowed stratification of patients with regards to subsequent response to metformin drug treatment, demonstrating the usefulness of plasma proteome profiling for precision medicine efforts“ is an overstatement and needs to be substantially toned down both in the abstract and the main text. Although it seems plausible that plasma proteome profiling will be useful for „precision medicine efforts“ the analyses shown in this article do not provide proof of this claim as the paper is lacking proper validation of results for predicting metformin responsiveness. Thorough validation, including validation in external, larger cohorts is needed.

A more comprehensive description of selection of the S3WP cohort is needed. Although a number of exclusion criteria are stated, it seems that there should have been a much larger number of potential participants not to be excluded for these reasons. How were they selected? Please provide details about the selection process with exact numbers for each exclusion/selection step. A flow diagram would be most helpful for transparent reporting of the selection process.

Minor points:

Reference 17 (submitted paper) should be included only if the paper was accepted.

Lines 258/259: replace „should be taking“ by „should be taken“

Reviewer #1:

Reviewer: Zhong et al use the latest Olink technology with a nucleic acid readout to profile ~1500 plasma proteins from their wellness cohort. This platform enables the probing of 522 proteins more than their recently published study that used the PCR readout. They extend their previous observations by finding more sex specific proteins and more genetic associations. They also have preliminary differences between T2D responders and nonresponders

The strengths of the study are:

- 1) The authors demonstrate that the new readout format is highly reproducible and similar to the PCR readout.
- 2) The increase the amount of sex specific proteins and loci associated with protein abundance
- 3) The preliminary study identifying a set of proteins associated in T2D associated with metformin responders and nonresponders.
- 4) The technical execution of the study is fine.

Authors: We are grateful for these kind words.

Reviewer: The weakness of the study are:

- 1) The authors did not invent the technology—they are simply using it as are many other labs.

Authors: It is correct that the authors did not invent the technology and the PEA has been published in hundreds of publications. However, this is the first publication using PEA combined with “next generation sequencing”. We have shown the power of this new technology in both wellness and T2D longitudinal studies and this makes it important for future precision medicine efforts using plasma protein profiling.

Reviewer: 2) There is no increase concepts from their very recently published wellness paper and a very modest increase in information.

Authors: The manuscript describes the vast amount of new information gained by using the new concept for plasma profiling. In addition, it describes for the first time the combination of PEA and next generation sequencing for plasma profiling.

Reviewer: 3) The T2D could be taken much further.

Authors: We agree and we have added new information regarding the T2D analysis and we have changed Figures 5 and 6, as well as the supplementary materials with a lot of additional data. We were originally planning to include this data into a separate follow-up paper, but we have decided to add it to the current manuscript as suggested by the reviewer. The Results and Discussion sections related to the new Figures and Tables have therefore been changed.

Reviewer #2:

Reviewer: The paper presents an impressive and relevant approach “next generation plasma profiling”, where large parts of the setup have been published previously. I will limit my comments to

the dry genetics data part reflecting the influence of genetic variations on plasma levels, as I am not fully able to assess the details in the experimental part. As far as I understand this slice of the workflow, the starting point is 30X WGS data. I do not really understand how the data are leveraged, or rather possibly not leveraged.

Authors: The manuscript describes the vast amount of new information gained by using the new concept for plasma profiling.

Reviewer: Following the NGS data generation the possibly exciting variants seem to be ignored and a suboptimal GWAS (n=101) for variants with MAF>5% is carried out. It is surprising why the GWAS made in the first place when 30X WGS data are available. The logic behind this should be explained better, or modifications should be made.

Authors: The reason to choose common variants with MAF > 5% for downstream analysis is due to the limitation of the sample size. In the wellness study, only 101 subjects were included. The statistical power for low-frequency and rare variants would be low and the multiple testing correction will be also poorly understood with less than 5 subjects carried the variants.

Reviewer: There are also no sign of using the genetics to adjust for population structure and cryptic relatedness that would be ideal here. As this aspects is not part of what is previously published it would be good to explain the logic behind this strategy.

Authors: This is a relevant comment. Before the GWAS, a mixed-effect modelling was performed to investigate the associations between protein expression levels and the population structure. No significant results were observed. So we didn't include it as a covariate in the analysis. We have added text in the Methods section to clarify this.

Reviewer #3:

Reviewer: This paper provides a thorough description of a very interesting novel approach for comprehensive and sensitive plasma profiling based on proximity extension assay combined with next generation sequencing. The manuscript is well structured and well written overall.

Authors: We are happy for these kind words.

Reviewer: Most of the methods and results are clearly presented and interpreted. However, I am concerned that the claim prominently expressed in the abstract and elsewhere that „The analysis also allowed stratification of patients with regards to subsequent response to metformin drug treatment, demonstrating the usefulness of plasma proteome profiling for precision medicine efforts“ is an overstatement and needs to be substantially toned down both in the abstract and the main text. Although it seems plausible that plasma proteome profiling will be useful for „precision medicine efforts“ the analyses shown in this article do not provide proof of this claim as the paper is lacking proper validation of results for predicting metformin responsiveness. Thorough validation, including validation in external, larger cohorts is needed.

Authors: This is a relevant comment. We have toned down the claims in the abstract as suggested by the reviewer. In the Discussion, we write: “... it is important to point out that more

in-depth analysis of these plasma protein profiles must be performed in larger disease cohorts to validate their use as clinical biomarkers.”

Reviewer: A more comprehensive description of selection of the S3WP cohort is needed. Although a number of exclusion criteria are stated, it seems that there should have been a much larger number of potential participants not to be excluded for these reasons. How were they selected? Please provide details about the selection process with exact numbers for each exclusion/selection step. A flow diagram would be most helpful for transparent reporting of the selection process.

Authors: A sentence of the inclusion criteria has been added to the Results section.

Reviewer: Reference 17 (submitted paper) should be included only if the paper was accepted.

Authors: This paper is now published and this reference has been updated.

Reviewer: Lines 258/259: replace „should be taking“ by „should be taken“

Authors: This has been corrected.

REVIEWERS' COMMENTS

Reviewer #2 (Remarks to the Author):

The paper has been improved, although the responses to some of the reviewer's comments are a bit defensive. The text changes are also quite limited here and there, but altogether a significant improvement in terms of understanding what was done. However, the concern in relation to the wording around metformin is not really addressed. The reviewer stated " ... is an overstatement and needs to be substantially toned down both in the abstract and the main text." The overstatement is still there, it is not sufficient with a standard remark on the need for further validation. Regarding novelty, I think it is not a problem that the Olink technology has been frequently used in earlier papers, I see it as a sign of solidity and not as a bug. The authors also combine with NGS and thus use it in a new manner. Nice to see the data availability text added.

We are grateful for the overall positive response by the reviewers on the revision of our paper by Zhong et al. entitled “Next generation plasma proteome profiling to monitor health and disease”. The comments and suggestions from the reviewer are relevant. We have therefore made a revision of the text.

Reviewer #2 (Remarks to the Author):

The paper has been improved, although the responses to some of the reviewer’s comments are a bit defensive. The text changes are also quite limited here and there, but altogether a significant improvement in terms of understanding what was done. However, the concern in relation to the wording around metformin is not really addressed. The reviewer stated “ ... is an overstatement and needs to be substantially toned down both in the abstract and the main text.” The overstatement is still there, it is not sufficient with a standard remark on the need for further validation. Regarding novelty, I think it is not a problem that the Olink technology has been frequently used in earlier papers, I see it as a sign of solidity and not as a bug. The authors also combine with NGS and thus use it in a new manner. Nice to see the data availability text added.

Authors: The statement regarding the stratification of the response to metformin treatment has been toned down in both abstract and the main text.

In addition, we have gone through the author checklist, reporting summary and policy summary, and answers all of the questions and issues raised by the editor. We have marked all changes in the manuscript by red color to highlight these.

We hope that the revised manuscript is suitable for Nature Communications.

Yours sincerely,

Mathias Uhlen

Mathias Uhlen, PhD., Professor
Science for Life Laboratory
Karolinska Institutet and Royal Institute of Technology, Stockholm, Sweden

Phone: +46 8 790 99 87 (secre)
e-mail: mathias.uhlen@scilifelab.se
Homepages : www.proteinatlas.org